# Noisy Neural Network Compression for Analog Storage Devices

**Berivan Isik**
Department of Electrical Engineering
Stanford University
berivan.isik@stanford.edu

**Kristy Choi**
Department of Computer Science
Stanford University
kechoi@cs.stanford.edu

**Xin Zheng**
Department of Electrical Engineering
Stanford University
xzheng3@stanford.edu

**H.-S. Philip Wong**
Department of Electrical Engineering
Stanford University
hspwong@stanford.edu

**Stefano Ermon**
Department of Computer Science
Stanford University
ermon@cs.stanford.ed

**Tsachy Weissman**
Department of Electrical Engineering
Stanford University
tsachy@stanford.edu

**Armin Alaghi**
Facebook
alaghi@fb.com

## Abstract

Efficient compression and storage of neural network (NN) parameters is critical for resource-constrained, downstream machine learning applications. Although several methods for NN compression have been developed, there has been considerably less work in the efficient storage of NN weights. While analog storage devices are promising alternatives to digital systems, the fact that they are noisy presents challenges for model compression as slight perturbations of the weights may significantly compromise the network's overall performance. In this work, we study an analog NVM array fabricated in hardware (Phase Change Memory (PCM)) and develop a variety of robust coding strategies for NN weights that work well in practice. We demonstrate the efficacy of our approach on MNIST and CIFAR-10 datasets for pruning and knowledge distillation.

## 1   Introduction

The rapidly growing size and large-scale training procedures of deep neural networks present new challenges in their storage, computation, and power consumption for resource-constrained devices (Dean et al., 2012; LeCun et al., 2015). Recent studies have demonstrated the promise of end-to-end analog memory systems for storing analog data, such as NN weights, as they have the potential to reach higher storage capacities than digital systems with a significantly lower coding complexity (Zarcone et al., 2018; Zheng et al., 2018). However, analog storage devices are noisy, which presents several key challenges for the compression task. First, the memory cell's noise characteristic is a non-linear function of the input value written onto the cell, and the range of input values must be limited due to constraints of the physics of device operation. Prior work models such memory cells as zero-mean white Gaussian channels when simulating the effect of the weight perturbations, and does not take real physical hardware device constraints into consideration (Upadhyaya et al., 2019; Zhou et al., 2020). Second, slight perturbations of the NN weights from the memory cell may cause the network's performance to plummet (Achille et al., 2019), which is not accounted for in most NN compression techniques (Han et al., 2016; Oktay et al., 2019).

34th Conference on Neural Information Processing Systems (NeurIPS 2020), Vancouver, Canada.

Motivated by the above challenges, we draw inspiration from classical information theory to develop a framework for encoding and decoding NN parameters to be stored on analog data storage devices (Shannon, 2001). In particular, our method: (a) leverages existing compression techniques such as pruning (Guo et al., 2016; Frankle & Carbin, 2019) and knowledge distillation (KD) (Hinton et al., 2015; Polino et al., 2018; Xie et al., 2020) to learn a compressed representation of the NN weights; and (b) utilizes various coding strategies to ensure robustness of the compressed network against storage noise. That is, we aim to minimize the performance degradation of the target network after it has been stored while simultaneously minimizing the number of memory cells used for model storage. For future work, we plan to combine parts (a) and (b) into a learned (robust) compression scheme.

Our experiments explore the effect of NN weight perturbations (due to the noisy analog storage devices) and ways to mitigate the effect of such perturbations on the compressed weights. Our results confirm the empirical findings of (Han et al., 2016; Frankle & Carbin, 2019), in that NN weights are robust to high levels of pruning. However, the remaining weights are sensitive to slight perturbations, and we show that: (a) protecting the sign of the parameters using extra one cell per parameter, (b) coding the parameters with small and large magnitudes separately, and (c) varying the amount of redundancy for weights with small and large magnitudes work well for achieving good classification performance with the reconstructed weights. As a concrete example, a naive baseline which achieves the baseline accuracy of a (pruned) noise-free ResNet-18 model uses an average of 1000 cells per weight for the coding procedure, while our strategy requires on average 1 cell per weight resulting in just a 0.5% loss in accuracy. In addition to pruning, we show favorable results for robust model training with knowledge distillation. Specifically, in both MNIST and CIFAR10, a smaller student network trained with noise injection in the weights is more robust to added noise at test time, relative to teacher and student networks trained without noise (Li et al., 2017; Xie et al., 2020).

We emphasize that our framework and coding strategies, though initially developed for analog storage, are more general – that is, they are broadly applicable to wireless communication settings of NN weights or other scenarios which subject the weights to additive noise.

## 2 Proposed Methodology

PCM is a strong candidate for analog-valued memory as its resistance levels (output) can be continuously tuned (Kuzum et al., 2012). For details regarding PCM cells and how the channel model in Figure 1 is built, we refer the reader to Appendix A. Although we can average over multiple cells when the channel response has high-variance, we aim to minimize this redundancy ($< 32$ digital memory cells per one floating point to outperform digital storage devices). That is, our objective is to store the compressed NN parameters on the minimum number of analog PCM cells, while preserving the network performance. To achieve this, we utilized a collection of techniques and evaluated the efficacy of each in the following section.

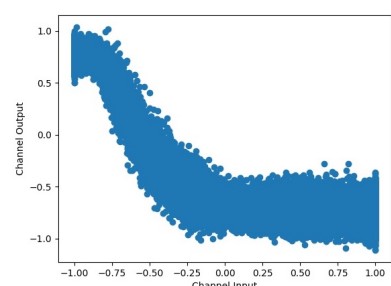

Figure 1: Characteristics of the channel noise in a PCM cell.

**Inverting the Channel:** The mean function of the PCM noise model $\mu : \mathcal{X} \to \mathbb{R}$ can be learned via a k-nearest neighbor classifier on the channel response measurements as shown in Figure 1. We learn an inverse function $f = \mu^{-1}$ which is used to remove the non-linearity in the mean, since $\phi = C \circ f$ is an identity function with zero-mean noise (Figure 2) where $C$ represents the channel, i.e. $\phi(x) = x + \epsilon$ where $\epsilon$ is noise. Thus the relationship between input weights $w_{in}$ to be stored and output weights $w_{out}$ to be read is:

$$w_{out} = \frac{\phi(\alpha w_{in} - \beta) + \beta}{\alpha}$$

where $\alpha$ and $\beta$ are scale and shift factors, respectively.

**Sign Protection:** When scaling the weights by $\alpha$ to fit them in range $[-0.65, 0.75]$ of $\phi$ (see Figure 2), small weights are mapped to values very close to zero. This is problematic because a majority of the trained NN weights have small magnitudes, and thus the NN with reconstructed

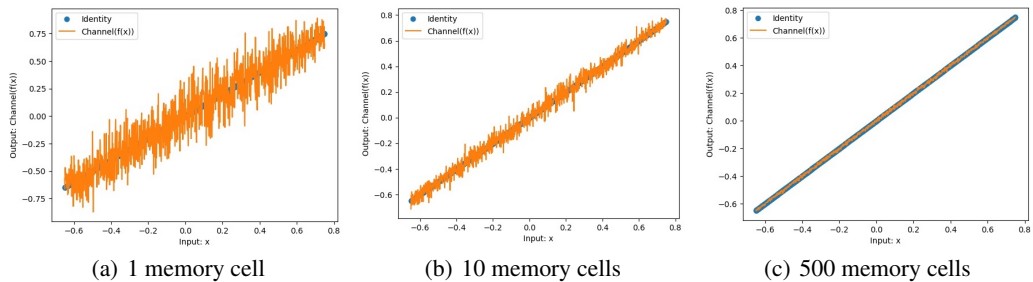

|     (a) 1 memory cell     |     (b) 10 memory cells     |     (c) 500 memory cells     |

Figure 2: Behavior of $\phi = C \circ f$ when outputs are read from an average over 1, 10, 500 cells.

weights will suffer a severe drop in performance due to sign errors (see Appendix D.1 for more details). Therefore, we store the sign and magnitude of the weights separately. The sign can be represented by 1 bit and can be stored using 1 cell. When we store magnitudes instead of actual weights, we can use an $\alpha$ that is twice as large, reducing the variance of noise because $\phi$ in Figure 2 is a noisy identity function that outputs:

$$w_{out} = \frac{\big((\alpha w_{in} - \beta) + \epsilon\big) + \beta}{\alpha} = w_{in} + \frac{\epsilon}{\alpha} \tag{1}$$

where $\epsilon$ is the noise of the device and $\frac{\epsilon}{\alpha}$ has smaller variance when $\alpha$ is larger.

**Adaptive Mapping:** Although protecting the sign bit leads to accuracy gains, we still require more than 32 devices to achieve the baseline accuracy (accuracy of the classifier without any additional noise), as shown in Table 1. Therefore, we use the observation that the majority of nonzero weights after pruning are quite small (see Appendix D.1). This implies that using different values of $\alpha$ depending on the magnitude of the weight (larger scale factor $\alpha$ for small weights) can reduce the overall variance of the cell's noise. This strategy requires us to use 1 extra bit to indicate whether a weight is small or large since two different $\alpha$'s are used in encoding and decoding. Together with the sign bit that we store separately, we need to store 2 bits per weight which can be done using 1 PCM cell per weight (Engel et al., 2014).

**Adaptive Redundancy:** As the last step, we propose to vary the number of devices used for larger and smaller weights. The average number of devices that we aim to minimize is:

$$r_{avg} = \frac{r_{small} \times N_{small} + r_{large} \times N_{large}}{N_{small} + N_{large}}.$$

where $N_{small}$ and $N_{large}$ are the number of small and large weights; $r_{small}$ and $r_{large}$ are the number of devices used per weight for small and large weights. Using more devices for larger weights (which are more critical for NN performance) increases the accuracy while it does not increase the average redundancy too much. As shown in Appendix D.1, most weights are sparse, e.g. $N_{small} = 2,209,074$ and $N_{large} = 5,615$ in pruned ResNet-18 trained on CIFAR-10.

We note that adaptive mapping and adaptive redundancy are similar to the adaptive coding strategies developed for task specific error correction codes (ECC). For instance, small and large inputs are treated differently for product operations in (Huang et al., 2015).

## 3  Experimental Results

For our experiments, we are interested in empirically investigating the following questions:

1. Can we learn a robust pruning scheme with the PCM array channel model?
2. Can we compress NNs via distillation in a way that is robust to channel noise?

For the details about datasets, models and hyperparameters used in the experiments, we refer the reader to Appendix B. In our experiments, we used the simulated channel model in Figure 1 that was built with the measurements taken from PCM arrays. We plan to store the weights on actual PCM arrays and read them back in the final version of this work.

### 3.1 Robust Pruning

In this setup, we leverage existing pruning techniques to assess whether we can learn to denoise using our simulated PCM array. For pruning, we consider the magnitude of all weights in a target network and zero out all weights smaller than that of the 80th percentile. We then apply channel noise to the non-pruned weights first with only inverse function $f$, then add sign protection, adaptive mapping, and adaptive redundancy. Our goal is to outperform binary storage devices, which can store a weight using 32 cells. Table 1 shows that without sign protection, we cannot do better than random prediction even with 30 cells. When we add sign protection, we see significant improvements; with adaptive coding, we obtain good results even with 2 cells per weight. Finally, when we add adaptive redundancy, we achieve the baseline accuracy 93.65%, with 5 cells. We also would like to emphasize that the accuracy loss is only 0.5% with 1 cell per weight on average. We refer the reader to Appendix D.1 for additional results on MNIST and experiments with Gaussian noise.

|  | 30 cells | 20 cells | 10 cells | 5 cells | 3 cells | 2 cells | 1 cell |
|---|---|---|---|---|---|---|---|
| Test acc. (no noise) | 93.65 | 93.65 | 93.65 | 93.65 | 93.65 | 93.65 | 93.65 |
| Test acc. (PCM noise+inv. channel) | 9.46 | 10.21 | 10.18 | 8.76 | 9.92 | 13.03 | 10.08 |
| Test acc. (+sign protection ) | 90.85 | 89.61 | 74.97 | 20.38 | 11.07 | 10.11 | 9.00 |
| Test acc. (+adaptive mapping) | 93.12 | 92.98 | 92.41 | 91.68 | 91.44 | 91.42 | 84.85 |
| Test acc. (+adaptive redundancy) | 93.45 | 92.98 | 92.96 | **93.65** | 93.05 | 93.32 | 93.11 |

Table 1: Accuracy of 80% pruned ResNet-18 on CIFAR10 when weights are perturbed by the PCM arrays channel model.

### 3.2 Robust Knowledge Distillation

In this experiment, we use knowledge distillation (KD) to compress a large target network (teacher) to a smaller network (student). We are interested in investigating whether training a student network with noise added to the weights will help make the network more robust to weight perturbations.

We consider various scenarios at test time. First, we evaluate the models in a noise-free setting. Then, we consider noisy settings where the same kind of noise is added to the weights as was applied to the student during training (+ noise). We also apply a slightly different type of white Gaussian noise (+diff noise) than presented during training. Finally, we consider different scenarios with the addition of the simulated noisy channel from the PCM array, coupled with the inverse mapping.

As shown in Table 2, the student network trained with noise (KD + noise) performs worse than the baseline teacher and student networks at test time in the noise-free setting. However, the "noisy student" outperforms all relevant baselines (including the teacher) in settings where different types of noise are added to the weights at test time. When the weights are slightly perturbed at test time, a compressed network trained with this method achieves up to 51% improvement in accuracy over baselines on CIFAR10. We refer the reader to Appendix D.2 for additional results on MNIST.

|  | Teacher | Student | KD | KD + noise (ours) |
|---|---|---|---|---|
| Test acc. (no noise) | 95.7 | 92.5 | 92.9 | **93.0** |
| Test acc. (+ noise) | 85.4 | 88.1 | 89.2 | **90.3** |
| Test acc. (+diff noise) | 10.5 | 51.0 | 68.4 | **76.5** |
| Test acc. (PCM noise+inv. channel) | 9.9 | 9.4 | 9.8 | 10.3 |
| Test acc. (+sign protection) | 40.5 | 11.0 | 10.6 | **15.6** |
| Test acc. (+adaptive mapping) | 94.4 | 69.0 | 62.0 | **76.6** |
| Test acc. (+adaptive redundancy) | 94.7 | 89.5 | 91.1 | **91.7** |

Table 2: KD results on CIFAR10 subject to additive Gaussian perturbations on the weights at test time. Our student network trained with noise (KD + noise) outperforms all relevant baselines, with the exception of the noise-free case. Number of cells per weight is 4 in PCM experiments.

# 4 Discussion & Conclusion

In summary, we proposed and explored several strategies for robustly compressing and encoding NN weights onto analog devices (PCM cells), and evaluated our method on pruning and knowledge distillation tasks for MNIST and CIFAR10. We note that in this work, we made the assumption of infinite precision when writing to and reading from memory cells with the aim to make full use of the storage capability of analog PCM devices. In practice, the complexity of read/write circuitry for high precision memory cells can be cost prohibitive. Limited precision, however, would compromise downstream classification accuracy of the reconstructed network. Our future work will take these practical limitations into consideration.

# 5 Acknowledgement

The authors would like to thank TSMC Corporate Research for technical discussions.

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

# Appendix

## A  Additional Details about PCM Cells

When programming the PCM cell with current pulses (input), the cell material would transition between low-conductive amorphous state and high-conductive poly-crystalline state. Various current pulses would generate different ratio between the two states, and thus put the cell into different resistance levels. By measuring typical output-input (resistance vs current) response from analog PCM array hardware, we build the PCM channel model as in Figure 1 (Zheng et al., 2018). The input and output have been scaled to be within [-1, 1] for the convenience of channel model. The output can be modelled as conditional Gaussian distribution with input-dependent means and variances. The variances are generated from multiple factors, including but not limited to device-to-device variation, cycle-to-cycle variation and read noise. Both the mean and variances are nonlinear functions of the input.

## B  Datasets, Models and Hyperparameters

**Datasets:**  For all our experiments, we consider two image datasets: the MNIST (LeCun, 1998) and CIFAR10 (Krizhevsky et al., 2009) datasets. We use the standard train/val/test splits for both, and for CIFAR10 we apply random horizontal flips as data augmentation during training.

**Models:**  For the experiments on MNIST, we use two architectures: (1) LeNet (LeCun et al., 1998) for the teacher network, and a 3-layer MLP (100 hidden units) for the student network. For CIFAR10, we use two types of ResNets (He et al., 2016): ResNet-18 for the teacher network (11.2M params) and a slim version of ResNet-20 (0.27M params) for the student network.

### B.1  MNIST:

We provide the architectural details and hyperparameters for LeNet and the MLP in Tables 3 and 4 respectively:

| Name | Component |
|---|---|
| conv1 | [$5 \times 5$ conv, 20 filters, stride 1], ReLU, $2 \times 2$ max pool |
| conv2 | [$5 \times 5$ conv, 50 filters, stride 1], ReLU, $2 \times 2$ max pool |
| Linear | Linear $800 \to 500$, ReLU |
| Output Layer | Linear $500 \to 10$ |

Table 3: LeNet convolutional architecture.

| Name | Component |
|---|---|
| Input Layer | Linear $784 \to 100$, ReLU |
| Hidden Layer | Linear $100 \to 100$, ReLU |
| Output Layer | Linear $100 \to 10$ |

Table 4: MLP architecture.

For both LeNet and the MLP, we use a batch size of 100 and train for 100 epochs, early stopping at the best accuracy. We use the Adam optimizer iwth learning rate = 0.001, and $\beta_1 = 0.9, \beta_2 = 0.999$ with weight decay = $5e^{-4}$. For KD, we use a temperature parameter of $T = 1.5$ and equally weight the contributions of the student network's cross entropy loss and the KD loss ($\alpha = 0.5$).

### B.2  CIFAR10:

We provide the architectural details and hyperparameters for the ResNet-18 and slim ResNet-20 used in our experiments in Tables 5 and 6 below:

For both ResNet-18 and slim ResNet-20, we use a batch size of 128 and train for 350 epochs, early stopping at the best accuracy. We use SGD with learning rate = 0.1, and momentum = 0.9 and weight decay = $5e^{-4}$. For KD, we use a temperature parameter of $T = 1.5$ and equally weight the contributions of the student network's cross entropy loss and the KD loss ($\alpha = 0.5$).

| Name | Component | |
|---|---|---|
| conv1 | $3 \times 3$ conv, 64 filters. stride 1, BatchNorm | |
| Residual Block 1 | $3 \times 3$ conv, 64 filters 
 $3 \times 3$ conv, 64 filters | $\times\, 2$ |
| Residual Block 2 | $3 \times 3$ conv, 128 filters 
 $3 \times 3$ conv, 128 filters | $\times\, 2$ |
| Residual Block 3 | $3 \times 3$ conv, 256 filters 
 $3 \times 3$ conv, 256 filters | $\times\, 2$ |
| Residual Block 4 | $3 \times 3$ conv, 512 filters 
 $3 \times 3$ conv, 512 filters | $\times\, 2$ |
| Output Layer | $4 \times 4$ average pool stride 1, fully-connected, softmax | |

Table 5: ResNet-18 architecture.

| Name | Component | |
|---|---|---|
| conv1 | $3 \times 3$ conv, 16 filters. stride 1, BatchNorm | |
| Residual Block 1 | $3 \times 3$ conv, 16 filters 
 $3 \times 3$ conv, 16 filters | $\times\, 2$ |
| Residual Block 2 | $3 \times 3$ conv, 32 filters 
 $3 \times 3$ conv, 32 filters | $\times\, 2$ |
| Residual Block 3 | $3 \times 3$ conv, 64 filters 
 $3 \times 3$ conv, 64 filters | $\times\, 2$ |
| Output Layer | $7 \times 7$ average pool stride 1, fully-connected, softmax | |

Table 6: Slim ResNet-20 architecture.

## C Additional Experimental Details

### C.1 Distillation

For MNIST, we train the noisy student with a $\mathcal{N}(0, 0.1)$ perturbation in the weights during training. At test time, "+diff noise" refers to a setup where $\mathcal{N}(0, 0.2)$ noise is added to the weights.

For CIFAR10, we train the noisy student network with a small $\mathcal{N}(0, 0.01)$ perturbation in the weights during training. At test time, "+diff noise" refers to a setup where $\mathcal{N}(0, 0.02)$ noise is added to the weights.

## D Additional Experimental Results

### D.1 Pruning

The sparsity of NN weights before and after pruning can be seen in Figure 3. In our joint pruning and storage scheme, we are storing only the non-pruned weights in the PCM cells. The indices of the non-pruned weights can be stored in digital storage devices since they are not analog values but integers from a finite codebook. After the pruning, $99.75\%$ of the non-pruned weights are categorized as small ($\leq 0.05$ in magnitude) while there are large weights with magnitude up to $0.6$. This makes it necessary to use different scale factors $\alpha$ for small and large weights to protect small weights ($99.75\%$ of the all weights stored in NVM cells) against noise.

We observe that our coding strategies are not limited to a particular analog storage device (PCM arrays). The adaptive protection strategy that we developed in this work improves NN performance against additive white Gaussian noise on the weights as well. As shown in Table 7, accuracy drops to $10.27\%$ (random prediction) when Gaussian noise with $0.05$ standard deviation is added without protection (without sign protection, adaptive mapping and adaptive redundancy). When our adaptive protection strategies are applied, accuracy ($93.55\%$) is comparable to the baseline accuracy ($93.65\%$) even with standard deviation $0.5$.

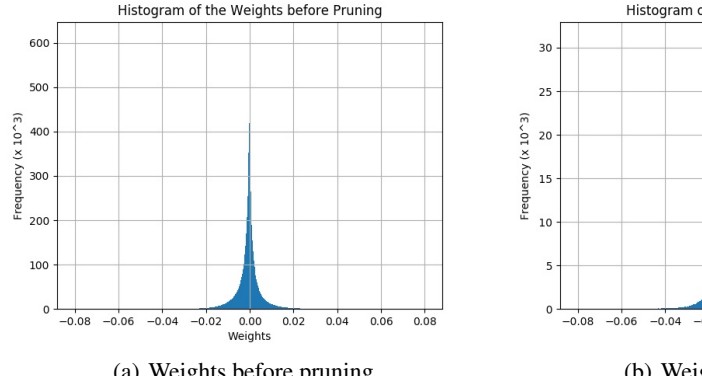

(a) Weights before pruning         (b) Weights after pruning

Figure 3: (a) Histogram of ResNet-18 weights trained on CIFAR10. (b) Histogram of ResNet-18 weights trained on CIFAR10, with 80% pruning.

|  | std=0.001 | std=0.005 | std=0.01 | std=0.05 | std=0.1 | std=0.5 | std=1 |
|---|---|---|---|---|---|---|---|
| Test acc. (+noise) without protection | 93.65 | 92.97 | 91.42 | 10.27 | 9.85 | 10.22 | 10.92 |
| Test acc. (+noise) with adaptive protection | 93.67 | 93.67 | 93.66 | 93.73 | 93.55 | 85.78 | 10.24 |

Table 7: Accuracy of $80\%$ pruned ResNet-18 on CIFAR10 when weights are perturbed by Gaussian noise. The baseline test accuracy without any added noise is 93.65.

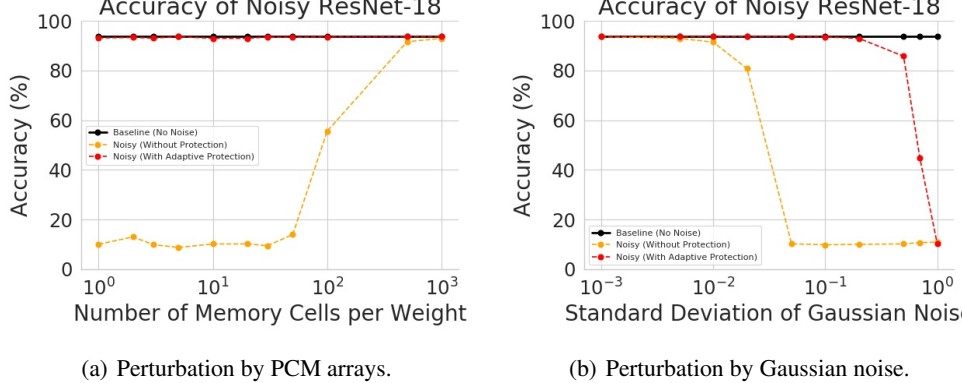

(a) Perturbation by PCM arrays.        (b) Perturbation by Gaussian noise.

Figure 4: Accuracy of $80\%$ pruned ResNet-18 on CIFAR10 when weights are perturbed by (a) PCM arrays, (b) Gaussian noise.

Figure 4 shows the effects of our coding strategies against both additive white Gaussian noise and a more realistic noise (added by PCM arrays).

|  | 30 cells | 20 cells | 10 cells | 5 cells | 3 cells |
|---|---|---|---|---|---|
| Test acc. (no noise) | 98.7 | 98.7 | 98.7 | 98.7 | 98.7 |
| Test acc. (PCM noise+inv. channel) | 10.32 | 10.3 | 10.32 | 10.32 | 10.32 |
| Test acc. (+sign protection ) | 98.7 | **98.74** | 98.66 | 98.71 | 98.39 |
| Test acc. (+adaptive mapping) | 98.75 | **98.77** | 98.72 | 98.81 | 98.65 |
| Test acc. (+adaptive redundancy) | 98.75 | 98.74 | **98.79** | 98.73 | 98.66 |

Table 8: Accuracy of $80\%$ pruned LeNet on MNIST when weights are perturbed by PCM arrays.

### D.2 Distillation

We give the distillation results on MNIST in Table 9. The student network trained with Gaussian noise (KD + Gaussian noise) outperforms all relevant baselines when Gaussian noise is added at test time. In addition to that, when channel noise is added at test time, the accuracy of noise-free network is achieved if all the strategies, i.e. inverse mapping, sign protection, adaptive mapping and adaptive redundancy, are applied. When channel noise is added, number of cells per weight is selected 4 in all experiments except (+adaptive redundancy). In (+adaptive redundancy) experiments, the average number of cells per weight is kept between 3 and 4.

| | Teacher | Student | KD | KD + Gaussian noise | KD + Channel noise |
|---|---|---|---|---|---|
| Test acc. (no noise) | 99.2 | 97.5 | **97.8** | 96.3 | 97.7 |
| Test acc. (+ noise) | 73.9 | 80.0 | 86.0 | **94.4** | 75.9 |
| Test acc. (+ diff noise) | 21.0 | 33.5 | 27.2 | **80.7** | 33.8 |
| Test acc. (PCM noise+inv. channel) | 11.6 | 8.3 | 10.1 | 9.8 | 9.8 |
| Test acc. (+sign protection) | 99.0 | 95.9 | 93.6 | 95.9 | **96.8** |
| Test acc. (+adaptive mapping) | 99.2 | 96.6 | 96.0 | 96.1 | **97.4** |
| Test acc. (+adaptive redundancy) | 99.2 | 97.3 | **97.7** | 96.3 | 97.6 |

Table 9: Knowledge distillation results on MNIST subject to additive Gaussian perturbations on the weights at test time. Our student network trained with noise outperforms most relevant baselines.

