# OpenReview forum: "Noisy Neural Network Compression for Analog Storage Devices"
_NeurIPS.cc/2020/Workshop/DL-IG — NeurIPSW 2020: DL-IG Poster_

### Official Review · AnonReviewer2 · 2020-10-30
**Hardware Neural Network Compression**

**Rating:** 7
**Confidence:** 3

**Review:**

This paper explores techniques for compression neural networks weights in a way that works well practically in a PCM.

I appreciate the applied application, I have to admit I didn't know what a PCM was before doing some external reading up on it. Would have helped reach a broader audience to include a bit of introductory text.

Otherwise seems like a reasonable set of things to consider for the specific hardware application in question, and I don't really feel too qualified or knowledgeable to judge beyond that.

---

### Official Review · AnonReviewer1 · 2020-11-07

**Rating:** 6
**Confidence:** 3

**Review:**

This paper demonstrates weight pruning and distillation for Phase Change Memory (PCM) using MNIST and CIFAR-10 datasets. The paper uses techniques to map the weights to the response of PCM, sign-bit protection and adaptive mapping for small/large weights.

This is an interesting paper, although its relevance to the present workshop is tenuous. This paper discusses analog storage mechanisms but if one were to think of analog computing, then it would be interesting to develop quantization schemes that  incorporate the path of the activations inside the network.

---

### Decision · Program_Chairs · 2020-11-07

Accept (Poster)